# Proteomic Analysis of Exosomes Secreted from Human Alpha-1 Antitrypsin Overexpressing Mesenchymal Stromal Cells

**DOI:** 10.3390/biology11010009

**Published:** 2021-12-21

**Authors:** Hua Wei, Erica Green, Lauren Ball, Hongkuan Fan, Jennifer Lee, Charlie Strange, Hongjun Wang

**Affiliations:** 1Departments of Surgery, Medical University of South Carolina, CRI 410, 173 Ashley Avenue, Charleston, SC 29425, USA; weihu@musc.edu (H.W.); greeeric@musc.edu (E.G.); 2Cell and Molecular Pharmacology and Experimental Therapeutics, Medical University of South Carolina, CRI 311, 173 Ashley Avenue, Charleston, SC 29425, USA; ballle@musc.edu; 3Pathology and Laboratory Medicine, Medical University of South Carolina, CRI 610, 173 Ashley Avenue, Charleston, SC 29425, USA; fanhong@musc.edu; 4Wellesley College, Wellesley, MA 02481, USA; jl121@wellesley.edu; 5Department of Medicine, Medical University of South Carolina, CSB 816, 96 Jonathan Lucas St., Charleston, SC 29425, USA; strangec@musc.edu; 6Center for Cellular Therapy, Medical University of South Carolina, Charleston, SC 29425, USA; 7Ralph H. Johnson VA Medical Center, Charleston, SC 29425, USA

**Keywords:** mesenchymal stromal cells, alpah-1 antitrypsin, exosome, extracellular vesicles, therapy

## Abstract

**Simple Summary:**

The overexpression of human alpha-1 antitrypsin (hAAT) in mesenchymal stromal cells (MSCs) improved their intrinsic properties with enhanced protective effects in previous murine models of type 1 diabetes. This study compared the protein profiles of extracellular vesicles (EVs) from bone marrow-derived control or hAAT-MSCs. EVs from both cell types share common features in size and exosome marker expression. By comparing common proteins in EVs from three donor cell lines using Gene Ontology (GO) analysis, we found that common EV proteins from all donors are important to cell adhesion and extracellular matrix organization. Differentially expressed proteins from hAAT-MSCs and MSCs are involved in cytokine signaling of the immune system, stem cell differentiation, and carbohydrate metabolism. This study shows that hAAT-MSCs have different profiles of paracrine effector exosomal proteins compared to control MSCs.

**Abstract:**

Extracellular vesicles (EVs) mediate many therapeutic effects of stem cells during cellular therapies. Bone marrow-derived mesenchymal stromal cells (BM-MSCs) were manufactured to overexpress the human antiprotease alpha-1 antitrypsin (hAAT) and studied to compare the EV production compared to lentivirus treated control MSCs. The goal of this study was to compare protein profiles in the EVs/exosomes of control and hAAT-MSCs using unbiased, high resolution liquid chromatography and mass spectrometry to explore differences. Nanoparticle tracking analysis (NTA) showed that the particle size of the EVs from control MSCs or hAAT-MSCs ranged from 30 to 200 nm. Both MSCs and hAAT-MSCs expressed exosome-associated proteins, including CD63, CD81, and CD9. hAAT-MSCs also expressed high levels of hAAT. We next performed proteomic analysis of EVs from three healthy donor cell lines. Exosomes collected from cell supernatant were classified by GO analysis which showed proteins important to cell adhesion and extracellular matrix organization. However, there were differences between exosomes from control MSCs and hAAT-MSCs in cytokine signaling of the immune system, stem cell differentiation, and carbohydrate metabolism (*p* < 0.05). These results show that hAAT-MSC exosomes contain a different profile of paracrine effectors with altered immune function, impacts on MSC stemness, differentiation, and prevention of cell apoptosis and survival that could contribute to improved therapeutic functions.

## 1. Introduction

Mesenchymal stromal cells (MSCs) are promising cellular therapy tools used in clinical trials. Based on their easy accessibility from multiple tissue sources, easy ex vivo expansion, and tissue protective and immunomodulatory properties [1,2], these cells have been studied for their paracrine effects on a number of human diseases [3,4]. Many factors, including donor age, cell expansion times, and cryopreservation, in clinical transplantation of MSCs can alter their secretory profiles and impact their therapeutic potency [1,5]. These factors are major barriers in the clinical translation of MSCs. Therefore, understanding the regulatory mechanisms in MSC secretory profiles would maximize their therapeutic potential.

Extracellular vesicles (EVs) are nanometer-sized membrane-bound vesicles secreted by all cells. They contain proteins, mRNAs, and microRNAs that can be delivered to recipient cells and function as “messengers” and modulators of recipient cell function [6]. EVs are generally divided into three types based on their biogenesis and biophysical characteristics: exosomes (40–150 nm), microvesicles (150–1000 nm), and apoptotic bodies (50–2000 nm) [7]. They can mediate the therapeutic effects of MSCs by modulating the immune system, inhibiting apoptosis and fibrosis, and promoting angiogenesis and cell growth under normal or pathological conditions [8,9,10]. Exosomes also carry a variety of cytoplasmic and membrane proteins such as receptors, enzymes, transcription factors, extracellular matrix proteins, nucleic acids, and lipids that impact function of neighboring cells [11]. 

In the world of cellular therapeutics, there is increasing evidence to suggest that MSC-derived exosomes may have advantages over infused MSCs based on their safety profile, low immunogenicity, and the ability to cross biological barriers [12,13]. Exosomes have good stability and permeability due to their surface phospholipid bilayer [14]. As nanoparticles, exosomes can penetrate the blood brain barrier and avoid potential occlusion of the pulmonary circulation related to infusion with MSCs [15]. Exosomes play a critical role in many pathways associated with various physiological and pathological functions [16]. As such, EVs may mediate many MSC-associated therapeutic effects, delivering therapeutic proteins and RNA to the recipient cells [17]. The functional protein components of EVs are at least partially responsible for the disease-modulating capacity of EVs derived from MSCs [18]. In fact, the therapeutic effects of cell free EVs have been increasingly reported for therapy of cardiac diseases, kidney diseases, and brain injury [19,20,21,22,23].

Human alpha-1 antitrypsin (hAAT) is a 52 kDa glycoprotein primarily produced in the liver [8]. *SERPINA1*, the gene encoding AAT, is transcribed from different promoters in hepatocytes and macrophages, respectively [24]. Stresses, such as inflammation, pregnancy, and infection, lead to increased circulating levels of AAT [25]. As a protease inhibitor, AAT protects tissues from degradation by destruction of a variety of proteases, especially neutrophil elastase. In our previous studies, human AAT-overexpressing MSCs (hAAT-MSCs) derived from bone marrow improved innate properties of human MSCs (hMSCs) with a higher proliferative capacity and faster migration. Infusion of hAAT-MSCs showed better efficacy in the prevention of onset of type 1 diabetes in spontaneous non-obese diabetic (NOD) mice compared with control MSCs [26]. Data from other laboratories also showed that hAAT-MSCs augmented the anti-inflammatory potential of MSCs and showed significantly better protection in graft-versus-host disease [27]. 

The goal of this study was to (1) dissect the components of exosome proteins harvested from lentivirus treated control- and hAAT- MSCs and (2) explain the potential mechanisms responsible for the improved protective effects. Bone marrow derived MSCs from three healthy donors were used to generate hAAT-MSCs and to harvest EVs for proteomic analysis. Differences in exosomal fractions were analyzed for functional differences in signaling pathways between hMSC or hAAT-MSCs. These data offer evidence supporting the potential of using EVs from hAAT-MSCs for therapy. Because of the lack of specific makers between different types of EVs [28], we interchangeably used the terms EVs and exosomes to refer to our cellular product throughout this paper.

## 2. Materials and Methods

### 2.1. Cell Isolation and Culture

MSCs were harvested from bone marrow specimens from three 19–25-year-old healthy African American male donors (D2, D3, D4) purchased from Stemexpress (Folsom, CA, USA). In brief, 10 mL of Ficoll-Hypaque gradient was added to bone marrow samples and centrifuged at 400× *g* for 30 min at room temperature. The lymphocyte layer was isolated, suspended in Dulbecco’s phosphate-buffered saline (DPBS), and centrifuged at 300× *g* for 5 min. The cell pellets were resuspended in alpha Minimum Essential Medium (α-MEM) supplemented with 20% fetal bovine serum (FBS), l-glutamine, and 1% penicillin/streptomycin. Cells were then seeded at a density of 0.5 × 10^6^ cells/cm^2^. Nonadherent cells were removed by washing with DPBS 3 days after, and cells were further cultured until they reached confluence. Cells were trypsinized and then sub-cultured in Dulbecco’s modified Eagle’s Medium (DMEM), 10% FBS, and 1% streptomycin and penicillin with densities of 5000 cells/cm^2^.

### 2.2. Viral Transduction

hMSCs were transduced with a lentivirus carrying pHAGE-CMV-a1aT-UBC-GFP-W to generate hAAT-overexpressing MSCs (hAAT-MSCs) as described previously [26,29]. Cells transfected with the control lentivirus, pHAGE-CMV-GFP-W (hMSCs), were used as controls [26,29]. In brief, MSCs at passages 2–3 were seeded at a density of 0.5 × 10^6^ cells per well in 6-well plates and cultured overnight at 37 °C with 5% CO_2_. Cells were then incubated with AAT or control viruses at a multiplicity of infection of 20 for 16 h before changing to fresh complete medium. Presence of GFP^+^ cells were detected under a fluorescent microscope 96 h after viral infection. Cells were then sorted by fluorescence-activated cell sorting (FACS) based on GFP expression to separate non-transfected cells.

### 2.3. Exosome Extraction

hMSCs or hAAT-MSCs at 70–80% of confluence were cultured in StemPro™ MSC SFM (Thermo Fisher, Waltham, MA, USA). Cell culture medium was collected 48 h later, centrifuged at 300× *g* for 4 min, and then at 2000× *g* for 10 min. Supernatant was collected and frozen at −80 °C for further use. Exosomes were harvested using ultracentrifugation at 10,000× *g* for 40 min at 4 °C. The collected supernatants were centrifuged again at 100,000× *g* for 90 min at 4 °C. The pellets were washed with PBS and then stored at −80 °C. The size distribution of exosomes was measured by ZetaView^®^ BASIC NTA (Particle Metrix, Mebane, NC, USA).

### 2.4. Protein Extraction and Western Blot

EVs stored at −80 °C were treated with protein lysis buffer containing protease cocktail (Sigma-Aldrich, St. Louis, MO, USA). Protein concentration was measured using a BCA Protein Assay kit (Thermo Scientific, Waltham, MA, USA). Hence, 20 µg of proteins were separated by SDS-PAGE polyacrylamide gel, transferred to Polyvinylidene fluoride (PVDF) membranes, and incubated with antibodies against CD63, CD9, CD81 (System Biosciences (SBI), Palo Alto, CA, USA), or human AAT (Cell signaling Technology, Danvers, MA, USA) separately, followed by incubation with the corresponding horseradish peroxidase-conjugated secondary antibodies (Cell Signaling Technology). Signals were visualized using a ChemiDocTM Imaging System (Bio-Rad, Hercules, CA, USA).

### 2.5. Sample Preparation for Liquid Chromatography and Mass Spectrometry

EV pellets were solubilized in 9 M urea and 50 mM Tris-HCL (pH8), reduced in 1 mM dithiothreitol and alkylated in 5.5 mM iodoacetamide. The urea concentration was reduced to 1.6 M with 50 mM ammonium bicarbonate. Proteins were digested with Lys-C at a 1:50 protease:protein ratio for 3 h at room temperature followed by trypsin digestion overnight at 37 °C at a 1:50 protease:protein ratio. The digestion was acidified with 1% formic acid and resulting peptides were desalted using C18 Stage Tips. The peptides were dried in a SpeedVac and stored at −80 °C.

### 2.6. Liquid Chromatography, Mass Spectrometry Data Acquisition Parameters

Peptides were analyzed by LC-MS/MS using an EASY nLC 1200 and Orbitrap Fusion Lumos Mass Spectrometer with instrument control software v. 4.2.28.14 (Thermo Scientific). Tryptic peptides were pressure loaded onto a C18 reversed phase Acclaim PepMap RSLC column, 75 µm × 50 cm (2 µm, 100 Å) (Thermo Fisher) and separated using a gradient of 5% to 40% B in 180 min (Solvent A: 5% acetonitrile/ 0.1% formic acid; Solvent B: 80% acetonitrile/ 0.1% formic acid) at a flow rate of 300 nL/min. 

High resolution (6 × 10^5^ ) FTMS survey scans were acquired with a mass range of *m*/*z* 375–1500. The automatic gain control target value was 4.0 × 10^5^ for the survey MS scans. Tandem mass spectra (MS/MS) were acquired on the most intense precursors with a cycle time of 3 s. Higher energy collisional dissociation was performed with a precursor isolation window of 1.6 *m*/*z*, a maximum injection time of 50 ms, and collision energy of 35%. Monoisotopic-precursor selection was set to “peptide”. Dynamic exclusion was enabled for 15 s to avoid redundant analysis of precursors within 10 ppm. Precursor ions with undetermined charge states, charge states of 1, or charge states greater than 5 were excluded. 

### 2.7. Data Analysis

For protein identification and quantification, the LC-MS/MS data was searched against the human SwissProt reviewed database (20,354 proteins, February 2020) and a database of common contaminants using the MaxQuant platform [30,31]. Data was processed in Perseus v1.6.14.0 [32]. Protein groups were filtered to remove those only identified by a modified peptide, matches to the reversed database, and potential contaminants. The raw protein intensities were log2 transformed and median normalized by subtracting the log2 median value. To avoid negative intensities, a constant value of 15 was added to all log2 intensities. Quantitative values were filtered to keep proteins that were quantified in at least two out of the three measurements for each group. Paired *t*-tests were performed and the threshold for change was a *p*-value < 0.05.

### 2.8. Bioinformatics Analysis

Identified proteins were compared with exosome data from the ExoCarta database (http://www.exocarta.org accessed on 26 August 2021). The logical relationship among the identified proteins was determined by Venny 2.1.0 (https://bioinfogp.cnb.csic.es/tools/venny/ accessed on 26 August 2021). The functional annotation of the proteins, including the Gene Ontology (GO) and the Kyoto Encyclopedia of Genes and Genomes (KEGG), was analyzed by DAVID Bioinformatics Resources 6.8 (https://david.ncifcrf.gov/ accessed on 3 August 2021).

## 3. Results

### 3.1. Generation and Characterization of hMSCs, hAAT-MSCs and Their Exosomes

MSCs from each donor (D2–D4) were infected with lentiviruses to overexpress hAAT. As shown previously [26], hAAT-MSCs and hMSCs shared similar fibroblast-like morphology under light microscopy (Figure 1A). GFP expression and secretion of hAAT in hAAT-MSCs have been measured here and previously (Figure 1A and [33]). The particle sizes of EVs from both hMSCs and hAAT-MSCs were within 30–200 nm, with most of them approximately 150-nm in diameter (Figure 1B).

Western blot analysis showed that tetraspanin exosome markers, CD63, CD81, and CD9, were expressed in exosomes from both hMSCs and hAAT-MSCs (Figure 1C, Appendix A). In addition, hAAT protein expression was much higher in hAAT-MSC-exsomes compared to hMSC-exosomes (Figure 1C, Appendix A).

### 3.2. Bioinformatic Analysis of Proteins in MSC-Exosomes from Individual Donors

A Venn diagram was used to analyze the common and different exosome proteins of hMSCs derived from three donors, D2, D3, and D4. There were 404, 382, and 365 proteins identified in exosomes from each donor respectively, with 316 common proteins present in all three samples (Figure 2A). Among the commonly expressed proteins, more than 85% of them matched with typical exosome proteins in the ExoCarta exosome database (Figure 2B). GO pathway enrichment was used to analyze the 316 common exosome proteins involved in the biological process, cellular component, and molecular function. Biological process analysis indicated that those proteins were enriched in cell adhesion, extracellular matrix organization, platelet degranulation, and protein stabilization (Figure 2C). Cellular component analysis showed that most of the proteins were typical for the extracellular exosome with signals prominent for focal adhesion and extracellular matrix. Molecular function analysis demonstrated that the proteins were enriched in cadherin binding involved in cell–cell adhesion, GTPase activity, GTP, protein, poly(A) RNA, actin filament, integrin and unfolded protein bindings and structural constituent of cytoskeleton (Figure 2C). KEGG pathway analysis showed that common exosome proteins were enriched in pathways involved in ECM–receptor interaction and focal adhesion (Figure 2D).

### 3.3. Bioinformatic Analysis of Proteins in hAAT-MSC-Exosomes from Individual Donors

Similar analysis of exosome proteins was performed in hAAT-MSCs from D2, D3, and D4. The Venn diagram indicated that there were 213, 276 and 296 exosome proteins in each sample, respectively. There were 178 common proteins among all three samples (Figure 3A). More than 85% of them matched with proteins typically seen in exosomes shown in ExoCarta (Figure 3B). GO pathway enrichment for biological process analysis indicated that the common proteins were enriched in chromatin silencing, cell adhesion, and extracellular matrix regulation. Cellular component analysis of the common proteins was typical for the extracellular exosome with signals prominent for the extracellular matrix. Molecular function analysis indicated that the common proteins were enriched in cadherin, integrin, and protein binding (Figure 3C). KEGG analysis showed that the common proteins were also involved in focal adhesion and ECM–receptor interactions (Figure 3D).

### 3.4. Comparation of Proteins from hMSCs- and hAAT-MSC-Exosomes by Horizontal Bioinformatic Analysis

Next, common exosomal proteins from three hMSC cell lines were compared with common exosomal proteins from hAAT-MSC cells. The Venn diagram revealed 170 commonly proteins among hMSC-exosomes and hAAT-MSC-exosomes (Figure 4A). Statistical analysis indicated that 61 of the 170 common proteins were significantly different between MSCs and hAAT-MSCs (Figure 4B,C, *p* < 0.05). Most of the differently expressed proteins were downregulated in hAAT-MSCs compared to hMSCs, including CAPZB, ANXA5, CD44, and many others (Figure 4C). There were also several upregulated proteins in hAAT-MSC exosomes compared to hMSCs including Antithrombin-III (encoded by *SERPINC1*), AAT (encoded by *SERPINA1*, Fructose-Bisphosphate B (encoded by *ALDOB*), tetraspanin 14 (*TSPAN14*), and Alpha-actinin-1 (encoded by *ACTN1*) (Figure 4C). Protein pathway analysis showed that the significantly changed proteins are involved in cytokine signaling in the immune system, maintenance of location in the cell, negative regulation of endopeptidase activity, leuokocyte cell–cell, stem cell differentiation, and metabolism of carbohydrates (Figure 4D).

## 4. Discussion

MSCs and their secretomes exert major protective functions in animal studies and clinical trials by affecting cell migration, proliferation, adhesion, and offering an anti-apoptosis microenvironment in vivo [9,34,35]. One important paracrine mechanism provided by MSCs involves the secretion of EVs that contain a robust profile of angiogenic and other effectors [36].

The overexpression of hAAT in MSCs improved the biological functions of native MSCs and led to better protection when given to non-obese diabetic (NOD) mice with new onset type 1 diabetes [26] or graft-verse-host-disease [27]. To further understand the potential mechanism of hAAT on the paracrine function of MSCs, we analyzed protein profiles in exosomes harvested from hMSCs and hAAT-MSCs. We found that exosomes from both hMSCs and hAAT-MSCs have comparable size and expression of exosome markers, including CD63, CD81, and CD9. hMSCs or hAAT-MSCs from three different donors were relatively homogeneous and shared most proteins. However, there were differences seen in the hAAT-MSCs compared to the control state with significant changes in several exosome proteins in addition to hAAT protein expression. The functions of these proteins involve alterations in cell migration, cytokine signaling of the immune system, stem cell differentiation, and carbohydrate metabolism. Our data indicate that the improved protective properties of hAAT-MSCs in vitro and ex vivo [26], found in our previous study, could result from their regulation of secreted exosome proteins of MSCs. These results may provide more clues about the functional improvement of hAAT-MSCs in therapy. Moreover, further evaluation of changed exosomal proteins in hAAT-MSCs may help identify critical signaling pathways or proteins involved in the regulation of MSC function and lead to a better understanding of the related mechanisms.

We found that exosomes from hAAT-MSCs had similar size distribution compared to hMSC-exosomes in this study. However, hAAT protein expression was increased in hAAT-MSC-exosomes compared to hMSC-exosomes. It is reasonable to postulate that the presence of AAT in exosomes could be responsible for the improved function observed in hAAT-MSCs and that hAAT-MSC-exosomes may have better therapeutic effects in disease therapy. Specific studies using hAAT containing exosomes will be required to determine therapeutic implications of this finding.

Our previous data indicates that hAAT-overexpressing cells have increased migration capacity compared to control MSCs [26]. The exosome protein profile supports the concept that exosomes may play a role in this process. We observed that proteins involved in cell adhesion and adherence junction, such as Destrin, Tubulin alpha chain, Annexin A, and Catenin delta, were significantly downregulated in hAAT-MSC-exosomes compared with hMSC-exosomes, suggesting hAAT-overexpressing exosomes might impact cell adhesion and cell migration by downregulating those proteins.

In addition, expression of proteins related to stem cell differentiation was also significantly decreased on hAAT-MSC-exosomes compared to controls. These include Annexin A6, Neuropilin-1, Catenin beta-1, and Endoglin. Our previous study also indicated that overexpression of hAAT enhanced the multilineage differentiation abilities of MSCs [26]. Thus, secreted AAT may be involved in MSC differentiation by regulating related proteins. Moreover, GO analysis revealed more proteins of hAAT-MSC-exosomes that are enriched in the negative regulation of cell apoptosis compared to hMSC-exosomes (Figure 2C and Figure 3C), suggesting that hAAT contained in exosomes may further reduce cell apoptosis as observed in some cell types (Gou, et al. unpublished data).

The intermediate metabolites in energy metabolism play an important role in shaping cellular functional properties [37]. Transplanted MSCs in control environments are usually present in a quiescent state in which cells appear to be primarily glycolytic [38,39], while the proliferation of MSCs increases in a nutrient-rich artificial culture environment and the proliferation of cells is dependent on oxidative phosphorylation [40,41]. In this study, KEGG analysis indicates that some proteins of hMSC-exosomes and hAAT-exosomes are involved in glycolysis /gluconeogenesis and carbohydrate metabolism. Aldolase B, a key enzyme in glucose and fructose metabolism, also increases in hAAT-MSC-exosomes compared to hMSC-exosomes. Aldolase B has been reported to promote cell proliferation in cancer metastasis [42,43]. The result in this study suggests AAT might be involved in MSC proliferation.

In this study, proteins involved in cytokine signaling in immune system, including CD44 antigen, Endoplasmin, Plastin-2, Fibronectin, Anastellin, Ug1-Y1, Ugl-Y2, Ugl-Y3, and Moesin, were significantly downregulated in hAAT-MSCs compared to MSCs (*p* < 0.05, Two-sided Student *t*-test). Moreover, the expression of proteins involved in immunoglobulin production, such as 60 kDa heat shock protein, Catenin beta-1, and Retinol-binding protein 4, also changed significantly in hAAT-MSC-exosomes compared to those of hMSC-exosomes. The results suggest AAT-containing EVs might regulate immune cell differentiation.

There are limitations to this study. Our data analysis was based on MSCs isolated from bone marrow samples of three donors. A larger number of donors and a validation cohort would provide more fidelity to the specific exosome protein changes. Our control cells also used a lentivirus construct that did not have hAAT included. Therefore, the extent of changes due to the pHAGE-CMV-GFP-W vector remains unknown. Further, the relevance of these findings to graft-versus-host-disease and type 1 diabetes animal models is currently speculative and can only be proven by direct experiments, since in vitro and in vivo exosome profiles may differ.

In summary, our results indicate that there are similarities as well as differences in exosomes from hMSCs or hAAT-MSCs from different donors. hAAT-MSCs have unique exosome profiles compared to hMSCs. The differently expressed proteins include those responsible for cell migration, stem cell differentiation, and immune modification. Next steps will focus on which exosomal proteins are important in human biology. Similar to the many biologic properties of hAAT, the exosomes from these hAAT overexpressing MSCs also target many biological systems.

## Figures and Tables

**Figure 1 biology-11-00009-f001:**
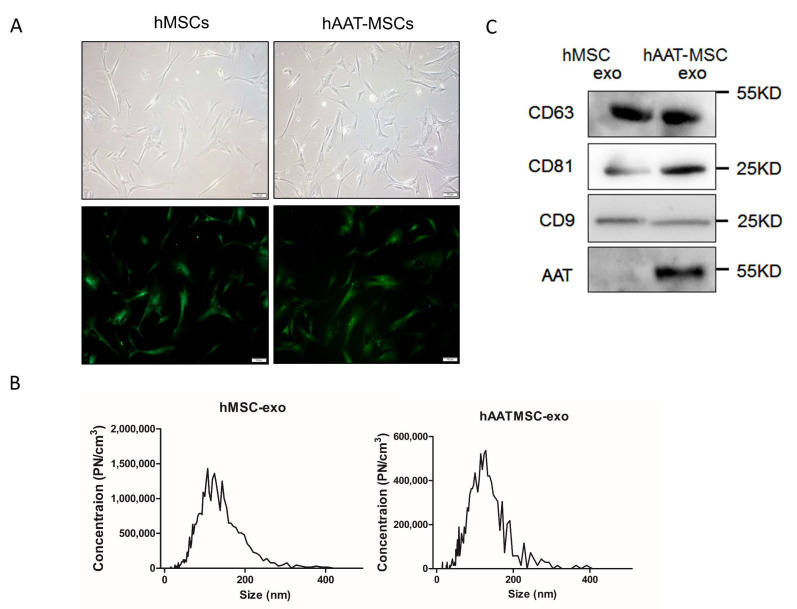
Characterization of hMSCs, hAAT-MSCs and their secreted exosomes. (**A**) Morphology of hMSCs and hAAT-MSCs in culture under the light microscope (upper panels) and after the cells were infected with hAAT or control-lentivirues, sorted by FACS and observed under fluorescent microsocope. Scale bar: 100 µm. Green indicates GFP+ cells. (**B**) Size distribution of exosomes derived from hMSCs and hAAT-MSCs measured by NTA. (**C**) Expression of CD63, CD81, CD9 and AAT, in hMSCs and hAAT-MSCs exosomes were detected by Western blot analysis.

**Figure 2 biology-11-00009-f002:**
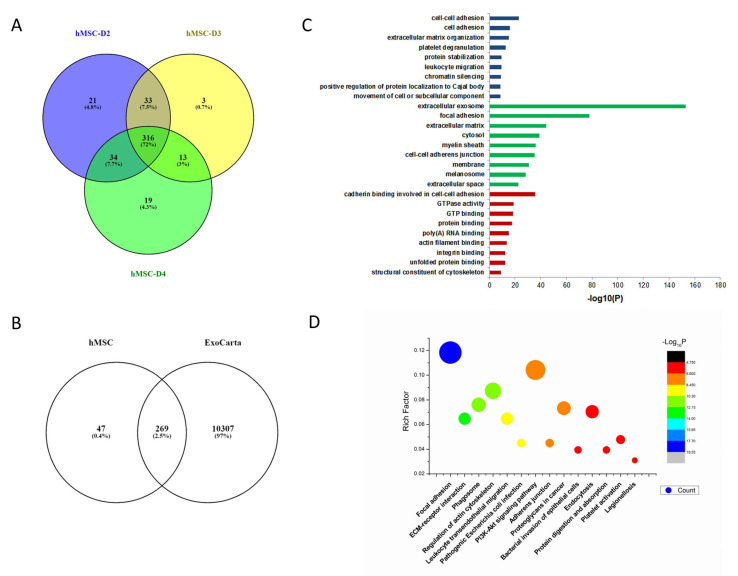
Bioinformatic analysis of hMSC exosomes. (**A**) Venn diagram of exosomes derived from hMSCs from bone marrow of three different donors. (**B**) Venn diagram of hMSC-derived exosomes against ExoCarta, an exosome database. (**C**) Gene Ontology analysis of hMSC exosomes. (**D**) KEGG analysis of hMSC exosomes.

**Figure 3 biology-11-00009-f003:**
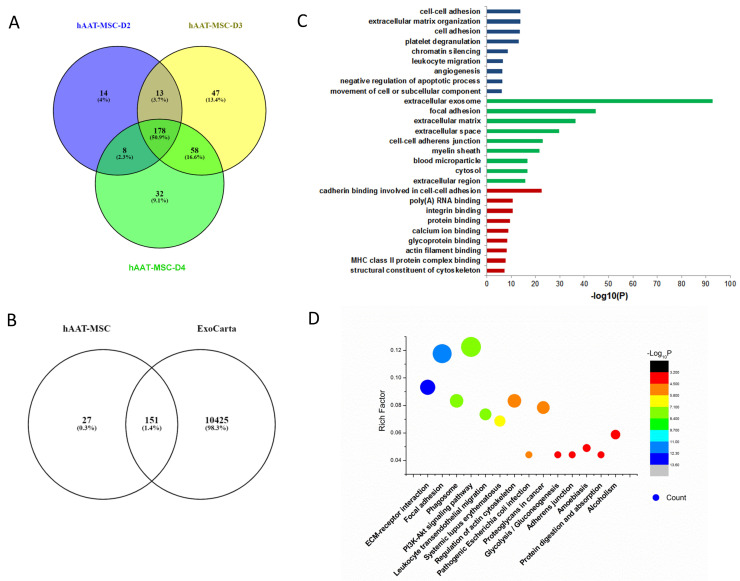
Bioinformatic analysis of hAAT-MSC exosomes. (**A**) Venn diagram of exosomes derived from three hAAT-MSCs cell lines. (**B**) Venn diagram of hAAT-MSC exosomes against ExoCarta. (**C**) Gene Ontology analysis of hAAT-MSC exosomes. (**D**) KEGG analysis of hAAT-MSC exosomes.

**Figure 4 biology-11-00009-f004:**
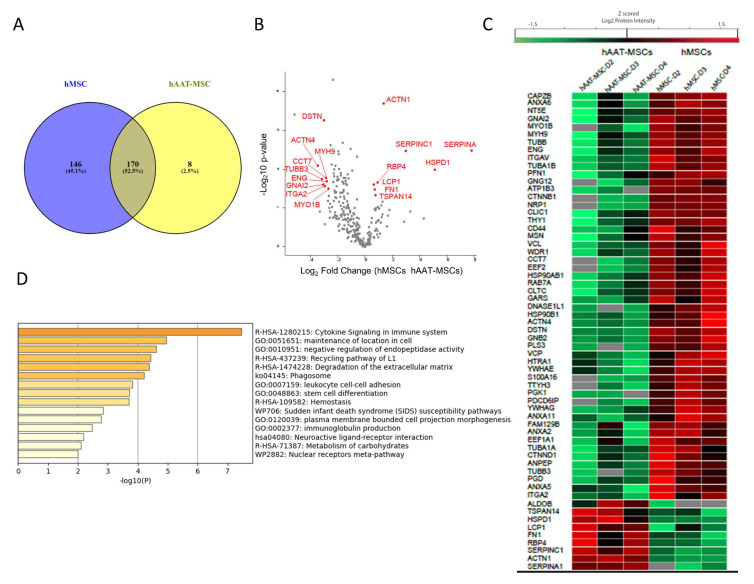
Horizontal bioinformatics analysis of exosomal proteins of hMSCs and hAAT-MSCs. (**A**) Venn diagrams of exosome proteins from hMSCs and hAAT-MSCs. (**B**) Volcano plot of differential proteins found in hMSC exosomes vs AAT-MSC exosomes. (**C**) Heat map of the shared exosomal proteins with significant differences (*p* < 0.05) in logarithmic Z score of the Protein intensity from hMSCs and hAAT-MSCs. Differences were compared by Student’s *t*-test. (**D**) Metascape functional enrichment of the shared proteins with significant difference from hMSCs and hATT-MSCs.

## Data Availability

The data presented in this study are available on request from the corresponding author.

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
