# Peer review of "Proteomic Analysis of Exosomes Secreted from Human Alpha-1 Antitrypsin Overexpressing Mesenchymal Stromal Cells"

_biology, 2021, doi:10.3390/biology11010009_

Round 1

Reviewer 1 Report

Hua et al have investigated the protein profiles of exosomes derived from hMSCs and hAAT-MSCs. Authors previously reported that hAAT-MSCs demonstrated improved biological functions in vitro and in vivo compared to hMSCs, thus warranting further investigations on possible mechanisms. The hypothesis for this study is that hAAT-MSC derived exosomes may exert these improved biological functions. Why choose AAT? The utility of these hAAT-MSCs (after lentiviral transduction) in therapy is questionable due to the risks associated with lentivirus.

  • The authors have used 3 donors to isolate hMSCs for their studies. This is not sufficient for subsequent profiling analysis with good statistical power.
  • The authors should also include control lentiviral transduction with GFP and without hAAT.
  • Figure 1 A – the images are not of good resolution. C – full size western blots should be provided, for all 3 donors and not just a representative sample.
  • The authors hypothesize that the improved biological functions of hAAT-MSCs could be attributed to the differences in exosomal protein cargo. They have performed bioinformatics analysis to show possible differences in the pathways. However, their observations of improved biological functions of hAAT-MSCs over hMSCs in vitro and in vivo cannot simply be attributed to differences in exosomal proteins based on their current bioinformatics analysis. There is no experimental evaluation of hAAT-MSC derived exosome function in their GVHD and T1D model to support their claims.
  • MSCs may release different exosome profiles in vivo compared to in vitro culture conditions and thus their current observations may not be relevant.
  • What are the future directions for these cells?

Author Response

Reviewer 1:

Q1: The hypothesis for this study is that hAAT-MSC derived exosomes may exert these improved biological functions. Why choose AAT?

Response: Many studies show that alpha 1 antitrypsin (AAT) has anti-inflammatory effects by suppressing cytokine production, complement activation, and immune cell infiltration. AAT also inhibits cell apoptosis and promotes cell proliferation. Injection of AAT to non-obese diabetic mice can reduce the intensity of insulitis, increase β cell mass, promote β cell regeneration, and prevent the onset of diabetes via modulating T regulatory cells. Evidence also indicates that infusion of AAT improves islet survival and function after transplantation. We chose AAT because we found that human AAT-overexpressing MSCs (hAAT-MSCs) derived from bone marrow improved innate properties of human MSCs (hMSCs) with a higher proliferative capacity and faster migration. Infusion of hAAT-MSCs showed better efficacy in the prevention of onset of type 1 diabetes in spontaneous non-obese diabetic (NOD) mice compared with control MSCs [24]. Data from other labs also showed that hAAT-MSCs augmented the anti-inflammatory potential of MSCs and showed significantly better protection in graft-verse-host disease [25]. We have enhanced the fourth paragraph of the introduction deals with the benefits of AAT.

Q2: The utility of these hAAT-MSCs (after lentiviral transduction) in therapy is questionable due to the risks associated with lentivirus.

Response: We agree with this reviewer that human trials may be problematic with this construct. We did reach out to the FDA concerning the regulations related to using hAAT-MSCs for therapy. The answer is that it is feasible but hAAT-MSC therapy must follow FDA regulations on both cell and gene therapies. The goal of this study is to explore the alternatives by using extracellular vesicles (EVs) as a cell-free therapeutic strategy. In this study, we compared protein profiles between exosomes derived from MSCs and AAT-MSCs. We also observed hAAT expression in EVs from hAAT-MSCs. This data lays the foundation for applying hAAT-MSC-derived exosomes in disease treatment.

Q3: The authors have used 3 donors to isolate hMSCs for their studies. This is not sufficient for subsequent profiling analysis with good statistical power.

Response: You are correct and we have added a limitations paragraph in the discussion.  To overcome the limitation of small sample size, we analyzed only the common proteins among three MSCs-derived exosomes and three hAAT-MSC exosomes by Venn diagram, and then comparatively analyzed the common proteins by Go analysis. This strategy helped avoid the potential inference from a diversity of different donors at some extent. In addition, this study only serves as a pilot study to test the hypothesis that proteins from hAAT-MSC-exosomes are different from those collected from MSCs. We will continue to do more experiments to further study significantly changed proteins screened by bioinformatics analysis.

Q4: The authors should also include control lentiviral transduction with GFP and without hAAT.

Response:  We apologize that this was not clear in our methods.  As stated in the material methods/viral transduction section, hMSCs transduced with the control vector, pHAGE-CMV-GFP-W, were used as controls throughout the study.  We have added a limitation that the analysis does not include a population of MSCs not treated with a lentivirus.

Q5: Figure 1 A – the images are not of good resolution. C – full size western blots should be provided, for all 3 donors and not just a representative sample.

Response: We replaced figure 1A with a high-resolution figure. We provided full-size Western blot photos to the journal. Western blots for the exosomes from the other two donors were included as supplemental data.

Q6: The authors hypothesize that the improved biological functions of hAAT-MSCs could be attributed to the differences in exosomal protein cargo. They have performed bioinformatics analysis to show possible differences in the pathways. However, their observations of improved biological functions of hAAT-MSCs over hMSCs in vitro and in vivo cannot simply be attributed to differences in exosomal proteins based on their current bioinformatics analysis. There is no experimental evaluation of hAAT-MSC derived exosome function in their GVHD and T1D model to support their claims.

Response: We agree with the reviewer and revised the tone of the manuscript as following:” Our data indicate that the improved protective properties of hAAT-MSCs in vitro and ex vivo, found in our previous study, could result from their regulation of secreted exosome proteins of MSCs.” We also have included a strong statement in the limitations paragraph that reads “Further, the relevance of these findings to GVHD and TID animal models is currently speculative and can only be proven by direct experiments.”

Q7: MSCs may release different exosome profiles in vivo compared to in vitro culture conditions and thus their current observations may not be relevant.

Response: We agree this could be a possibility. This is also added to the limitations.

Q8: What are the future directions for these cells?

Response: We will conduct more studies on these cells with the goal to apply hAAT-MSCs and/or their released EVs in therapy.

Reviewer 2 Report

The manuscript entitled “Proteomic Analysis of Exosomes Secreted from Human Alpha‐1 Antitrypsin Overexpressing Mesenchymal Stromal Cells” by Dr. De Hua and colleagues reports on a protein profile comparative analysis on extracellular vesicles harvested from three donor bone derived mesenchymal stromal cells (MSCs) and alpha‐1 antitrypsin‐overexpressing‐MSCs. Results indicate that both MSCs and hAAT‐MSCs expressed exosome‐associated proteins such as CD63, CD81 and CD9, while, as expected, hAAT‐MSCs also expressed high levels of hAAT. The results of the present analysis improve our knowledge on paracrine characteristics of MSCs. These data therefore present possible therapeutic applications.  Although the experimental design is limited (only three cell donors were enrolled), the ms is well written and well organized. 

Thank you for letting me work as reviewer for this work. I have a several observations

General comments
•    The results of the present investigation are interesting. However, the experimental design is very poor as only three donor MSCs cell lines were analyzed. The authors should justify this choice.
•    The clinical application of the findings of the present study should be highlighted in the discussion section
•    Although included in the abstract as main results of the study, CD63, CD81 and CD9 are not mentioned in the discussion section, while, contrariwise, the findings on CD44 are discussed but not included in the abstract

Minor
Lines 17, 112 please remove the bold style
Line 24 “(p < 0.05) These” please include the period
Lines 63-65 AAT is delivered in an inducible manner from the liver and blood cells upon activation of Serine Protein inhibitor-A1 (SERPINA1) inflammation-responsive promoter PMID: 3500042.  Moreover, AAT protein levels increase during inflammation, whisht rising also in normal physiological conditions, such as pregnancy (PMID: 33015055). Both notions should be included in the introductive section as a background, as underling the anti inflammatory/protective effect of the hAAT‐MSCs in vitro model described in this study
Line 64 please remove the underling under “tissues”
Line 259 please include the supporting reference. Citation n.24?

Author Response

Reviewer 2:

Q1: The results of the present investigation are interesting. However, the experimental design is very poor as only three donor MSCs cell lines were analyzed. The authors should justify this choice.

Response: Please see the response above. We have added the small sample size to a paragraph on limitations.

Q2: The clinical application of the findings of the present study should be highlighted in the discussion section.

Response: Thanks for the suggestion. We added the discussion about the clinical application in the manuscript.

Q3: Although included in the abstract as main results of the study, CD63, CD81, and CD9 are not mentioned in the discussion section, while, contrariwise, the findings on CD44 are discussed but not included in the abstract

Response: Thanks for the comment. CD63, CD81 and CD9 are markers to identify exosomes. We have added this information in the discussion section to indicate that MSCs and hAAT-MSCs have a similar exosome component. In addition, we found that eight proteins including CD44 involved in cytokine signaling in the immune system were changed significantly in hAAT-MSCs compared to MSCs. These results suggest the role of AAT in the regulation of immuno-modification function. We now include this information in the revised abstract.

Q4: Lines 17, 112 please remove the bold style.

Response: This was corrected.

Q5: Line 24 “(p < 0.05) These” please include the period

Response: This was corrected.

Q6: Lines 63-65 AAT is delivered in an inducible manner from the liver and blood cells upon activation of Serine Protein inhibitor-A1 (SERPINA1) inflammation-responsive promoter PMID: 3500042.  Moreover, AAT protein levels increase during inflammation, whisht rising also in normal physiological conditions, such as pregnancy (PMID: 33015055). Both notions should be included in the introductive section as a background, as underling the anti inflammatory/protective effect of the hAATMSCs in vitro model described in this study

Response: Thanks for the comment. We have added these two references.

Q7: Line 64 please remove the underling under “tissues”

Response: This was corrected.

Q8: Line 259 please include the supporting reference. Citation n.24?

Response: This was corrected.

Reviewer 3 Report

n this manuscript, the author demonstrates the " Proteomic Analysis of Exosomes Secreted from Human Alpha‐1 2 Antitrypsin Overexpressing Mesenchymal Stromal Cells". The manuscript is very interesting and well written. I have some recommendations that need to be addressed:

  1. Figure 1 quality is very poor needs to be changed with the representative image.
  2. Line 179 D81 should be CD81.
  3. Figure4 B quality is poor. it should be changed.
  4. Line 297 " were significantly changed in hAAT-MSC" should be written as Significantly downregulated
  5. The manuscript does not explain the mechanisms.  

Author Response

Reviewer 3:

In this manuscript, the author demonstrates the " Proteomic Analysis of Exosomes Secreted from Human Alpha‐1 Antitrypsin Overexpressing Mesenchymal Stromal Cells". The manuscript is very interesting and well written. I have some recommendations that need to be addressed:

Q1. Figure 1 quality is very poor needs to be changed with the representative image.

Response: High-quality figure has been uploaded.

Q2. Line 179 D81 should be CD81.

Response: This was corrected.

Q3. Figure4 B quality is poor. it should be changed.

Response: High-quality figure has been uploaded.

Q4. Line 297 " were significantly changed in hAAT-MSC" should be written as Significantly downregulated.

Response: This has been corrected.

Q5. The manuscript does not explain the mechanisms.

Response: We agree with the reviewer. As suggested in the title, the goal of this study is to compare the difference in proteins between hAAT-MSCs and control MSCs. Further studies out of the scope of this manuscript are needed to explain the mechanisms.

Round 2

Reviewer 1 Report

NA